# Digital Intentions in the Fingers: I Know What You Are Doing with Your Smartphone

**DOI:** 10.3390/brainsci13101418

**Published:** 2023-10-06

**Authors:** Laila Craighero, Umberto Granziol, Luisa Sartori

**Affiliations:** 1Department of Neuroscience and Rehabilitation, University of Ferrara, via Fossato di Mortara 19, 44121 Ferrara, Italy; 2Department of General Psychology, University of Padova, 35131 Padova, Italy; umberto.granziol@unipd.it (U.G.); luisa.sartori@unipd.it (L.S.)

**Keywords:** action observation, action recognition, digital intentions, digital affordances, kinematics

## Abstract

Every day, we make thousands of finger movements on the touchscreen of our smartphones. The same movements might be directed at various distal goals. We can type “What is the weather in Rome?” in Google to acquire information from a weather site, or we may type it on WhatsApp to decide whether to visit Rome with a friend. In this study, we show that by watching an agent’s typing hands, an observer can infer whether the agent is typing on the smartphone to obtain information or to share it with others. The probability of answering correctly varies with age and typing style. According to embodied cognition, we propose that the recognition process relies on detecting subtle differences in the agent’s movement, a skill that grows with sensorimotor competence. We expect that this preliminary work will serve as a starting point for further research on sensorimotor representations of digital actions.

## 1. Introduction

The history of the Internet goes back a few decades; email has been around since the 1960s, and file sharing since at least the 1970s. But it was the creation of the World Wide Web (WWW) in 1989 that revolutionized the history of communication. By now, it is no longer possible to imagine a world without the Internet. Today, there are five billion Internet users worldwide, which is 63 percent of the global population. Of this total, 4.65 billion are social media users [1]. Most users claim that the smartphone is the electronic device they use the most [2], which consists of a portable device that combines mobile and computing functions in a single unit. Most smartphones feature thin, slate-like form factors with large, capacitive screens to support multi-touch gestures. Tapping, swiping, and scrolling are some of the new hand actions required to reach new goals, such as opening a file, browsing multiple photos, and shifting a page.

Determining how this digital progress affects cognitive processes is one of the main challenges for contemporary behavioral neuroscience. Technological inventions, which include any tool, method, or skill designed to facilitate our daily functions, cause alterations in our behaviors and can profoundly affect our cognitive systems. Throughout our evolutionary history, our brains have been reshaped by the advent of tool making and usage [3], language [4], writing, and arithmetic systems [5]. Although the spread of touchscreen devices, such as smartphones and tablets, dates back to around 2010, there is little literature on the influence that the acquisition of digital skills and the constant use of technological devices is having on sensorimotor abilities and related cognitive processes. It is essential to note that a direct comparison between different generational cohorts is as necessary and topical as ever. In fact, only in this specific historical era is it possible to compare the effects of a technological invention on such abilities and processes in people born before and after its disclosure, as the so-called Digital Immigrants and Digital Natives [6] currently coexist. This coexistence will disappear within a few decades.

The theory of embodied cognition argues that the mind, the body, and their environment are highly interrelated and, therefore, mutually dependent on each other [7,8]. According to this approach, human cognition is deeply rooted in the interactions of the body with its physical environment [9]. Affordances are the sensory characteristics of an object that intuitively imply its use and function, and change depending on the intentions or goals of the individual. This principle emerges from the long-known evidence that the distal goal of the action influences movement execution [10]. When grasping a bottle, the positioning of the fingers depends on the intention to pour, move, throw [11,12], show [13], or pass it to somebody [14]. Sensorimotor representations are formed with the repeated experience of interacting with the world and are shared among those able to reach the same goal [15]. This sharing allows observers to be sensitive to early differences in visual kinematics and to use them to discriminate between movements performed with different intentions [16,17]. This ability increases as the motor ability to replicate the observed movement increases, as suggested by consistent results found in children [18], professional athletes [19], dancers [20], and musicians [21]. Basketball athletes, for example, can discriminate earlier and more accurately successful and erroneous throws than individuals with comparable visual experience (e.g., sports journalists). They achieve this goal by reading the body’s kinematics, and they are very good readers due to their personal motor experience [19]. How this ability developed in everyday typing actions on smartphones, however, has never been investigated before.

Typing text on a smartphone touchscreen is widely used to perform various tasks, such as email, Internet browsing, texting, and social media [22]. As is the case in the physical space [23,24,25,26,27,28,29], our online activities can be broadly classified into two categories: perceive or act. The terms used to categorize these different online behaviors are, respectively, “content consumption” and “content generation” [30,31]. Content consumption refers to the act of reading, listening, and viewing various forms of digital media. Examples of apps that involve this type of activity are those that allow access to web browsers (e.g., Google, Firefox, Safari), to weather information (e.g., 1Weather, Weather Now, AccuWeather), and news (e.g., Apple News, Google News, The Week, Flipboard). Content generation, instead, describes the various practices that result in any type of digital content, including text and voice messages, video files, photos, etc., to be shared with the digital community via blogs, email apps, and social media sites (e.g., Facebook, WhatsApp, Instagram, Twitter). As in the physical space, the main difference between the two categories relies on the consequences of the action. In the physical space, when you push an object (i.e., to act), the object changes position. Conversely, when you look at an object (i.e., to perceive), the object remains as it is. Only during content generation app use does the action have an effect on the web or others. For example, you can write the same question on WhatsApp or on Google, e.g., “what will the weather be like on Tuesday?”. The question written on a WhatsApp chat will determine a consequence in the receivers (i.e., to act): they may be happy because you are planning a trip together or angry because they have an exam that day and cannot join the group. The question written on Google, on the contrary, gives access to information already present and does not modify it (i.e., to perceive): it is like reading a newspaper. Thus, typing text on a smartphone touchscreen may be executed either to generate or to use digital content. Interestingly, evidence is present on the influence of functional characteristics of digital actions on the involvement of the sensorimotor system. Specifically, results have shown that patterns of inter-touch intervals were different for content consumption and content generation, and that the presence of recent greater activity in content generation influenced somatosensory cortical activity [30]. Moreover, evidence indicated that digital space is coded according to the same rules that apply in the physical space [23,24,25,26,27,28,29]. That is, peripersonal space representation is activated when app icons address digital actions able to generate content, and extrapersonal space representation is activated when they address digital actions just able to perceive content [31]. These findings suggest that digital affordances change based on an individual’s intentions or goals, just like in the physical world.

Currently, no study has tested whether an observer is able to recognize the agent’s intentions by exploiting differences in the interaction with digital affordances, as suggested by the theories of embodied cognition. Also, according to this approach, expertise should play a role during action observation by allowing high levels of accuracy in action detection. Unfortunately, there are currently no data to evaluate the ability to type on a touchscreen. The only information available concerns a study on typing speed, commonly considered a measure of typing ability on a keyboard. In detail, a large-scale dataset on mobile text entry collected via a web-based transcription task [32] performed by 37,370 volunteers (age range from 10 to 60 years) revealed the impact of demographic factors and typing styles on performance. Specifically, the number of written Words Per Minute (WPM) decreased with age. The highest typing speed was found in young people between 10 and 19 years of age, with an average of 39.6 WPM, while older people between 50 and 59 years of age showed an average of 26.3 WPM. Regarding typing styles, participants who reported using two fingers (bimanual action) were significantly faster than those who used only one finger (one hand action).

Since there was essentially no literature on the subject, to start investigating the relationship between action and cognition in the digital space, we chose a very simple experimental approach with few stimuli but involving many participants. Specifically, to verify if observers are able to decode the agent’s digital intention, we enrolled a Digital Native [6] (19 years old, writing with both thumbs, WPM = 41.3 [33]) and we video recorded her finger movements on her smartphone keyboard while texting a question to a friend on WhatsApp (i.e., content generation; “Gen1” video) and while typing the same question on Google (i.e., content consumption; “Con1” video). A different question was typed on both WhatsApp (“Gen2” video) and on YouTube (“Con2” video). This allowed us to create two pairs of videos (GenCon1 and GenCon2). In each pair, the person typed the same sentence with different intentions. The videos showed the hand seen from the side (so as not to show the screen) while typing the phrase, but not during app selection. We then performed a large-scale experiment online on the Qualtrics platform (Provo, UT). The required task was to observe each pair of videos (Gen vs. Con) and indicate in which of the two videos the agent was chatting on WhatsApp. In addition to the questions related to demographics, we also asked for the typing style (i.e., single or double hands, index finger/s, thumb/s, or other).

We hypothesized that the ability to recognize movements performed with different intentions is also present when we observe someone typing on their smartphone. We reasoned, based on the parallelism with the physical world, that the recognition process should be rooted in the capacity to detect very subtle differences in the agent’s typing. We expected the recognition process to increase with decreasing age, given the greater typing ability in younger subjects [32], which, according to embodied cognition, should increase the accuracy in detecting differences in the observed digital action.

## 2. Materials and Methods

### 2.1. Participants

Five hundred and nine (274 F, 235 M; age_mean_ = 31.84; age_sd_ = 14.65) individuals completed the task. Participation in the task was anonymous and voluntary. As soon as the participant voluntarily clicked on the link to participate in the online experiment, ethical and privacy information were provided, and the following sentence was presented: “By continuing to fill out the questionnaire, I express my consent to participate in the research”. Data collection was carried out from January to March 2022. Procedures were in accordance with the guidelines of the 1964 Declaration of Helsinki. The project has been approved by the Ethical Committee for the Psychological Research of the University of Padova (Protocol n. 4602). No information about the participants’ identity (neither name nor email address) was recorded.

### 2.2. Stimuli and Procedure

Stimuli consisted in videos showing, from a side view, the hands of a 19-year-old female agent while typing on a smartphone with two thumbs simultaneously. The agent was unaware of the experimental hypothesis. The position of the hands and the smartphone was kept constant in all videos. As the experimental hypothesis required to show an ecological stimulus, the agent used her personal smartphone to perform well-consolidated action patterns. For the same reason, when using WhatsApp, she was asked to actually converse with a friend and to insert the pre-determined sentence in a way that was consistent with the speech. This chat has been around for years, and conversations were frequent every day regardless of this study. Finally, to know the correspondence between the movement of the fingers and the result on the keyboard, the Screen Recorder was activated in the agent’s smartphone.

The stimuli consisted of two pairs of videos without audio. The Italian translation of the phrase “what will the weather be in Padua on Tuesday” (i.e., “che tempo farà a Padova martedì”) was used in the GenCon1 pair of videos. The Italian translation of the phrase “how do you put on mascara” (i.e., “come ti metti il mascara”) was used in the GenCon2 pair of videos.

In both Gen1 (Gen1 stimulus) and Gen2 (Gen2 stimulus) videos, the agent wrote the sentence on WhatsApp while chatting with her friend. In video Con1 (Con1 stimulus), the phrase was written in the Google search window. In video Con2 (Con2 stimulus), the phrase was written in the YouTube search window. The videos simply showed the finger movements required to input the phrase; they did not show the actions required to choose the app. All the videos are present as Appendix A (Appendix A: Gen1 stimulus; Appendix A: Con1 stimulus; Appendix A: Gen2 stimulus; Appendix A: Con2 stimulus).

The task was generated with Qualtrics platform, promoted via social media, and self-administered. The link could only be accessed once from each account. Participants’ information was also collected about gender (M/F/Non-binary), year of birth, and typing posture: (i) with two thumbs, (ii) with the index (by pressing the single letters), (iii) with the thumb (by pressing the single letters), and (iv) with the index finger (by swiping on the letters for each word).

The instructions were as follows: “You will be presented with 2 pairs of videos where a girl writes on the phone. In a video she chats on WhatsApp. In the other she writes the same sentence on Google or YouTube. After watching both videos you will be asked if the girl is chatting on WhatsApp in video A or in video B. You can review the videos as many times as you want. With the back arrow you can go back to video A even after having seen video B. Good fun!”

After presentation of the instructions, the videos of the first pair were presented one at a time, and the subjects could review each video as many times as they wanted. When the subjects decided to continue, the following question was presented: “Guess in which video the girl is chatting on WhatsApp: A or B?”. To answer, they had to tick the choice. Then, the same procedure was used for the second pair of videos. Both the order of the pairs and the order of the videos were randomized.

We chose a forced-choice task, considering it the most appropriate since the participants did not have explicit choice cues and had to rely on unconscious effects in their decision making [34]. Moreover, this protocol allowed us to have a single question for the two pairs of videos, given that in both pairs the generative app was WhatsApp, while the consumption app varied.

## 3. Data Analysis and Results

The analysis did not include the individuals who typed using swiping because their numbers were too low (*n* = 19). The resulting sample was of four hundred and ninety people (264 F, 226 M; age_mean_ = 31.74; age_sd_ = 14.72). Participant data for those who typed using only one index finger (*n* = 78) and one thumb (*n* = 82) were collapsed (one hand action). As a consequence, a variable named typing was created, containing two levels reflecting the habits of typing: both thumbs (*n* = 330; 176 F, 154 M age_mean_ = 25.58; age_sd_ = 9) and one finger (*n* = 160; 88 F, 72 M; age_mean_ = 44.44; age_sd_ = 16.04). A *t*-test for independent sample by groups (both thumbs vs. one finger) showed a statistically significant difference for age (*t*_(488)_ = −16.62, *p* < 0.0001), indicating that the mean age was lower for those participants who claim to use both thumbs (M age_mean_ = 25.58; age_sd_ = 9; one finger age_mean_ = 44.44; age_sd_ = 16.04) to type on the smartphone.

Given the binary nature of the task, generalized mixed effects models (GLMMs) were tested considering the Video pair (GenCon1 vs. GenCon2), the typing modality (both thumbs vs. one finger), and age (as continuous variable) as fixed factors. To increase the readability of the results, we centered all the predictors. The user ID was used as clustering variable and random factor. All the GLMMs were tested by using the lme4 package [35] on R software [36]. A significant effect of age (*β* = −0.02, *p* < 0.01, OR = 0.98 [0.97–0.93]) was found. An increase in age by one year was associated with a decrease in the odds of selecting the correct response by a factor of 0.98. Such an effect was dependent from the typing modality, as indicated by a statistically significant two-way interaction: age × typing (*β* = −0.03, *p* = *0*.02, OR = 0.97 [0.94–0.99]). The simple slope analysis showed that an increase in age by one year was associated with a decrease in the odds of selecting the correct response only for participants who typed using both thumbs (*β* = −0.04, *p* < 0.001, OR = 0.96). In fact, the slope of participants who typed with one finger was not statistically significant (*β* = −0.002, *p* = 0.72, OR = 0.99) (Figure 1). The results showed a difference between video pairs (*β* = 0.85, *p* < 0.001, OR = 2.35 [1.64–3.37]). The odds of correctly responding to the task in pair GenCon1 were 2.35 times higher than the odds of correctly responding to the task in pair GenCon2. The other effects emerged as non-statistically significant (see Table 1 for the overall model; in bold statistically significant results).

The same model was re-tested on the Subsample (*n* = 288; 144 female, 144 male), where typing modality was paired and matched across the sexes (72 both thumbs F, 72 one finger F, 72 both thumbs M, 72 one finger M). All results were coherent (See Table 1 for the Subsample model). A *t*-test for independent samples by groups also showed a statistically significant difference for age (*t*_(286)_ = 11.73, *p* < 0.0001) in the Subsample. The mean age was higher for those participants who claimed to use one finger (age_mean_ = 43.97; age_sd_ = 16) than for those using both thumbs (age_mean_ = 25.87; age_sd_ = 9.38) to type on a smartphone.

In order to better understand the interaction between age and typing on the odds of correctly responding to the task, we transformed the continuous variable age into a discrete variable. We based this transformation on the most common way to categorize Internet users, introduced by Prensky in 2001 [6]: digital Immigrants (DI; born before 1980) versus Digital Natives (DN; born after 1980). Accordingly, the overall sample was divided into DN (352, age_mean_ = 23.47; age_sd_ = 5.23) and DI (138, age_mean_ = 52.85; age_sd_ = 8.90). We tested GLMMs: the Video pair (GenCon1 vs. GenCon2), the typing modality (both thumbs vs. one finger), and the group (DN vs. DI) were used as fixed factors. The results were coherent with the previous analysis. The odds of correctly responding to the task of DI participants were 0.64 times lower compared to the DN counterpart (group: *β* = −0.45, *p* = 0.02, OR = 0.64 [0.44–0.93]). The two-way interaction group × typing was significant (*β* = −1.01, *p* < 0.01, OR = 0.36 [0.17–0.77]). Post hoc tests showed that when typing with both thumbs, the odds of correctly responding to the task was lower in DI than in DN (*p* = 0.0057). When typing with one finger, there was no difference between DI and DN (*p* = 0.99). Interestingly, DN typing with both thumbs did not differ from DN (*p* = 0.48) and DI (*p* = 0.48) typing with one finger (Figure 2). The results showed a difference between video pairs (*β* = 0.91, *p* < 0.001, OR = 2.48 [1.72–3.59]). The odds of correctly responding to the task in pair GenCon1 were 2.48 times higher than the odds of correctly responding to the task in pair GenCon2. Also, for this analysis, the same GLMM model was re-tested on the Subsample, where DN (179, age_mean_ = 23.85; age_sd_ = 5.69) and DI (109, age_mean_ = 53.10; age_sd_ = 9.11) were matched on sex and typing modality. All results were confirmed.

### 3.1. Discussion of Behavioral Results

The results indicated that the ability to respond correctly varies with age and style of typing on a smartphone. Specifically, the participants’ ability to respond correctly declined with age, but only if they declared that they type with two thumbs. On the contrary, if they declared to type with one finger (one thumb or one index finger), their ability to respond correctly did not vary with age. Precisely, only Digital Immigrants (DIs—born before 1980), compared to Digital Natives (DNs—born after 1980) [6], who declared to type with both thumbs performed worse than the others (see Figure 2). Thus, the expected result of higher guessing ability in younger individuals was confirmed, but only when the stated typing style was the one most frequently used by younger individuals. This latter information derives from the statistical analyses obtained from our sample, which showed a significantly younger age for those who usually write with two thumbs compared to those who write with only one finger (regardless of the finger used).

The expected results were based on the evidence that in the physical world, the ability to discriminate between movements performed with different intentions depends on the sensitivity to early differences in visual kinematics [16,17]. Thus, on the videos presented as stimuli, we performed both a descriptive analysis of the typing sequence and a kinematic analysis of the agent’s finger movements and compared the results between each video of the pair to identify which movement hints could be used by the observer to make the discrimination task. Since the video pair GenCon1 was easier to guess than GenCon2, we expected greater differences between the content generation and content consumption stimuli of the first pair.

#### 3.1.1. Descriptive Analysis of Typing Sequence in Videos

Each video used as experimental stimulus was synchronized with the video of the agent’s smartphone screen while typing (all the videos are present as Appendix A; Appendix A: Gen1 hand and screen; Appendix A: Con1 hand and screen; Appendix A: Gen2 hand and screen; Appendix A: Con2 hand and screen). The frame-by-frame viewing of each video made it possible to identify the movements of the fingers and the relative keys pressed. Figure 3 schematically illustrates the typing sequence for each thumb presented in the GenCon1 and GenCon2 videos.

In the GenCon1 pair, the presence of the accent selection tool for Google required selecting two accents (“farà” and “martedì”) in the Con1 video, which the autocorrect function in WhatsApp (Gen1 video) made unnecessary for the first accent (A′). In the Gen1 video, the autocorrect function required first pressing the function key and then the cued word for the second accent (I′) (the symbol ↑_text_ in Figure 3 indicates these actions). The accent selection tool requires swiping over the possibilities to select the choice. Also, thanks to the autocorrect function, the first letter of the word Padova (city name) was automatically entered as a capital letter on the Gen1 video (WhatsApp), while for the Con1 video (Google), it was necessary to select the caps key (the symbol ↑ in the Figure 3 indicates this action).

In the GenCon2 pair, we compared WhatsApp with YouTube where the software features were the same. However, given the ecological way we used for recording the videos, differences were still present due to the agent’s spontaneous writing. Specifically, the first ‘T’ letter of the phrase in the Con2 video was pressed by the left thumb, while in the Gen2 video, by the right thumb. Moreover, the agent erroneously wrote “maacara” instead of “mascara” in the Con2 video. This required the selection of the cued correct word at the end of the writing (the word _mascara_ in the Figure 3 indicates the action). These differences, however, are independent of the app’s specific features.

#### 3.1.2. Kinematic Analysis of the Agent’s Finger Movements

Since the data were obtained from a single subject, and given the low number of observations, it was not possible to apply the same analysis used in the only study that attempted to compare typing differences between content generation and consumption [30]. Therefore, we applied the same principles used in the analysis of reaching actions in the physical world, which consider the movement from the starting position to the target position. Specifically, we defined the “basic typing action” as the path the finger takes from when it releases a key (defined here as the starting position) to when it presses the next key (i.e., the target position) [37].

The software used to synchronize and analyze the stimuli was Kinovea^®^ software (Kinovea; Version 0.8.15; Kinovea open source project, www.kinovea.org (accessed on 15 April 2022)) [38,39]. From plain video-recordings of movements, the software allows for measuring distances and times manually or using semi-automated tracking to follow points and check live values or trajectories.

For each finger (i.e., left and right thumbs), Kinovea^®^ was used to locate the marker that was placed at the border between the nail and the skin at the point of greatest contrast. We used semi-automated tracking, that is, Kinovea^®^ kept track of the marker movement for the whole video, while the experimenter only adjusted it frame-by-frame when necessary.

Each video used as experimental stimulus was synchronized with the video of the agent’s smartphone screen while typing. The synchronization took place by linking the frame in which the finger touched the first key and the frame in which the first letter of the sentence appeared on the touchscreen keyboard. Synchronized videos made it possible to identify the basic typing actions performed by each finger (see videos: Gen1 hand and screen, Con1 hand and screen, Gen2 hand and screen, Con2 hand and screen). The path of the finger from the frame corresponding to the appearance of the first letter (i.e., starting position) to the frame of appearance of the second letter (i.e., target–object location) has been analyzed and considered a basic typing action.

The kinematic analysis was performed separately for each finger involved in typing, and only basic typing actions that started at the same starting key and ended on the same target key were compared.

For each basic typing action, we considered the speed data exported from Kinovea^®^. The speed data of each basic typing action performed by each thumb in each condition were compared through a distribution-free Overlapping Index proposed by Pastore and Calcagnì [40]. The Overlapping Index evaluates the overlapping area among two kernel density distributions. The higher the index, the more the overlapping and, consequently, the similarity between different distributions (for the sake of simplicity, such an index will be expressed as a percentage). Such analysis has been made by using the overlapping package [41] in R.

To further investigate this effect, we computed a series of parameters classically measured in reaching tasks [13,16]: (i) Peak Velocity (the maximum value in the velocity profile), (ii) Peak Deceleration (the minimum value in the acceleration profile), (iii) Time to Peak Velocity (the time to the maximum value in the velocity profile, normalized with respect to the reaching time), and (iv) Time to Peak Deceleration (the time to the minimum value in the acceleration profile, normalized with respect to the reaching time). The parameters of each basic typing action performed by each thumb in the same video pair were compared through a Tau_U index. This index is used to measure the amount of non-overlap between two trends of scores. It varies between −1 and 1. Zero indicates the presence of overlap (i.e., the trends are overpowered), while 1 (in absolute value) indicates the presence of non-overlap. A z-test was associated with this index to test its significance. With *p* < 0.05, the distributions are different because there is a statistically significant non-overlap. We excluded GenCon2 left thumb data from the analysis due to the too low number of basic typing actions considered.

For the sentence GenCon1, we considered nine basic typing actions for the left thumb and eighteen basic typing actions for the right thumb. For the sentence GenCon2, we considered five basic typing actions for the left thumb and eleven basic typing actions for the right thumb.

Data relative to the Overlapping Index indicated that the percentage of overlap was lower than 50% between basic typing actions executed with the right thumb in the GenCon1 videos (mean = 48%, SD = 15, range: 18–75). It was over 60% for actions executed with the left thumb in the same videos (mean = 66% SD = 16, range: 35–85) and for actions executed with the right (mean = 62% SD = 16, range: 31–83) and left thumbs in the GenCon2 videos (mean = 66% SD = 24, range: 27–92) (Figure 4).

The results relative to the Tau_U index of the kinematics parameters of each basic typing action performed by each thumb in the same video pair showed that the only non-overlapping parameter was Peak Velocity for the right thumb in the GenCon1 pair of videos (*p* = 0.046), indicating a higher Peak Velocity for the Gen1 video (180 m/s^2^) than for the Con1 video (123 m/s^2^).

These findings suggest that differences in kinematics are more evident for the GenCon1 than for the GenCon2 video pair. In GenCon1, for actions performed with the right thumb, the mean percentage of overlap for speed was 48%. This low percentage of overlap appears to be mainly due to a difference in Peak Velocity, which was significantly greater when the agent used WhatsApp compared to Google.

### 3.2. Discussion of Typing Sequence and Kinematic Results

Expected results were confirmed, showing greater differences between the content generation and content consumption stimuli of the GenCon1 pair.

Considering the typing sequence, the main difference between the two pairs of videos was that, while in GenCon1, the accent selection tool was present on the content consumption stimulus (Con = Google) and not on the content generation stimulus (Gen = WhatsApp), in GenCon2, this tool was never present (Con = YouTube; Gen = WhatsApp). It is worth noting that the accent selection tool requires swiping over the possibilities to select the choice, an action present twice in the GenCon1 content consumption stimulus. It is possible that only experts can recognize the swipe in the stimulus and know that it is a digital affordance that is absent in WhatsApp. For GenCon2, the only differences between the stimuli were the switch in the use of the two thumbs when typing the “T” letter and a spontaneous error by the agent who typed “A” instead of “S” (adjacent keys on the keyboard) in the content consumption stimulus. Differences in GenCon2, however, do not depend on the specific affordances of the app used and, therefore, cannot be used as clues for the discrimination of the different typing intentions (i.e., to discriminate the app used).

As regards the kinematic parameters, it was found that the only non-overlapping parameter was Peak Velocity for the right thumb in the GenCon1 pair of videos. In particular, the Peak Velocity was higher in the content generation stimulus (i.e., WhatsApp; 180 m/s^2^) than in the content consumption stimulus (i.e., Google; 123 m/s^2^). This result is in agreement with the literature, which shows that the movements performed for the “communicative” condition (similar to chatting on WhatsApp) are characterized by a kinematic pattern different from that obtained for the “individual” condition (similar to finding information on Google) [13].

It is worth noting that the lower number of typing actions identified for the right thumb in pair GenCon2 (*n* = 11) than in pair GenCon1 (*n* = 18) may have influenced the results, preventing the difference in Peak Velocity from emerging. However, these findings suggest that the differences in kinematics were more pronounced for GenCon1 than for the GenCon2 video pair.

## 4. General Discussion

Nearly two decades after the large-scale deployment of smartphones began, their use permeates almost every moment of our day. Many actions that used to be performed by interacting with physical objects (e.g., flipping through the pages of a newspaper) are now performed through a touchscreen (e.g., scrolling down). The literature on embodied cognition has consistently shown that observers are able to easily recognize the actions of others when directed towards physical objects. In particular, evidence suggests that observers are able to distinguish between movements carried out with different intentions using early kinematic differences [16,17]. Furthermore, research has shown that this competence improves as the ability to reproduce the observed movement increases [18,19,20,21].

The experimental question that inspired the present study was whether the ability to recognize movements performed with different intentions is also present when we observe someone typing on their smartphone, and whether this ability increases with the ability to type. As in the physical world, there are two broad categories of activities that can be carried out with the smartphone: perceive and act. These online behaviors are categorized under the terms “Content consumption” and “Content generation”, respectively [30,31]. The former denotes acts undertaken to consume digital content, which leave the environment unaffected. The second describes the acts that lead to the sharing of any type of digital content, influencing the reactions or behaviors of those who would use this content. Therefore, to test the hypothesis, we showed participants two videos in which the hands of a young girl typed the same sentence while actually chatting with a friend (Content generation) and while searching for that information on the web (Content consumption). The participants had to choose the video in which, according to them, the girl was chatting. Two pairs of videos were presented. WhatsApp was the content generation app employed in each of them. Google was the content consumption app in one pair while YouTube was the consumption app in the other.

The main result of this study indicated that the ability to respond correctly varies with age and style of typing on the smartphone. Specifically, only DIs who declared that they type with both thumbs—i.e., the typing style mostly utilized by DNs, and to a lesser extent by DIs—performed worse than the others. Furthermore, the analysis of the typing sequence and the kinematics of the agent’s finger movements indicated that there were greater differences between the Gen video and the Con video of the pair with the highest probability of discrimination. This suggests that each cohort has its own typing style. Those who habitually use that typing style behave as more expert observers, guessing more, probably using both the presence of the swipe and the differences in movement speed as clues to perform the task.

We predicted that the ability to guess would increase with decreasing age, because research on the ability to recognize actions in the physical world suggests that it grows with increasing motor competence [18,19,20,21] and because it is generally accepted that younger people are more adept at using smartphones. Therefore, in agreement with our hypothesis, we would like to argue that DI participants who type with two thumbs are less adept at smartphone interaction. Unfortunately, there are currently no data to evaluate the ability to type on a touchscreen. The only information currently available concerns a study on typing speed [32], commonly considered an indication of typing ability. The results of that study showed that the number of written Words Per Minute (WPM) decreases with age, and that those who report using two thumbs are significantly faster than those who use only one finger. Unluckily, the study did not consider the interaction between age and typing, which would have been helpful in interpreting the present results. As well, we do not know the typing speed of our participants, only their typing preference.

Thus, we can only discuss possible reasons why DI participants who type with two thumbs might be less proficient in touchscreen interaction than the other groups. The most likely cause concerns the difference in the stages of approaching typing on a touchscreen between the various generational cohorts; while the younger ones have started typing on modern touchscreens, the older ones had to adapt to a long evolutionary history of digital innovations. In fact, older people started typing on a typewriter and then on a computer, where thumbs were used (if at all) solely to tap the space bar. The habitual use of typewriters or computers may have developed the Hunt-and-Peck typing skill, a technique that uses one finger of each hand to press keys on the keyboard. It is conceivable, therefore, that most of them started typing on the 3 × 4 alphanumeric keyboard of their cell phones using only one finger to write SMS (Short Message Service) messages with a push-button device. To enter a character, it was required to press the designated key and to tap multiple times to cycle through the character options for that key. In the mid-2000s, almost all smartphones had a built-in QWERTY keyboard with separate keys for each character, which drastically reduced the size of each key to fit a typewriter-sized keyboard in a portable device. Also, within a decade, physical keyboards were replaced by digital touch keyboards, which are much more sensitive to touch than their predecessors. Later, larger smartphones came to the market that necessarily required the use of both hands to reach each key. Plausibly, some people continued to use the type of typing already established with experience, that is, holding the phone with one hand and using the finger of the other hand to type, while others tried to adapt to the new way of interacting with the touchscreen, that is, holding the phone with two hands and using both thumbs to type. Basically, the new type of smartphone interaction requires reversing the functional role of the thumb and forefinger: the thumb becomes the active effector (instead of being just an opposing finger), and the forefinger is used together with the other fingers to stabilize the phone. It is known that sensorimotor skill acquisition determines long-term changes in the finger areas of the primary motor cortex [42]. Therefore, it is possible that in young people, the neuronal representation of the thumb reflects its function as an effector because the sensorimotor practice of touchscreen typing is initiated (usually) at an early age. On the contrary, in the elderly, the extensive time spent using the thumb primarily as an opposing finger may impede or make more challenging the neuroplasticity process required to codify the new function of the thumb. The different functional representation at the cortical level could, therefore, determine a difference in the ability to effectively use the thumb as an active effector while typing. To check for this possibility, the authors of the present work have already planned a single-pulse Transcranial Magnetic Stimulation (TMS) study for mapping motor cortex representation to identify the neuroplasticity underpinning digital interaction, its correlation with age and style of typing, and, possibly, with typing skill.

According to the current findings, there was no difference in typing action recognition between DNs typing with two thumbs, DNs typing with one finger, and DIs typing with one finger. Given that the videos showed the agent using two thumbs to type, this suggests that recognition of the typing action does not require the observer to use the same typing style as the agent. This result agrees with evidence coming from monkey [43,44,45,46] and human [47,48,49,50,51] studies showing that the sensorimotor system does not encode the movements but the goal of the action [52]. For instance, the same cerebral activation occurred in people when watching the same action carried out by a human hand, a robot hand, or a tool [50], that is, regardless of the observed effector. Furthermore, the same findings lead us to argue that DNs typing with two thumbs, DNs typing with one finger, and DIs typing with one finger have the same level of typing ability. Again, we have no data to support this interpretation and can only discuss possible causes. As for DNs, we can assume that the decision to type in a certain way (two thumbs or one finger) was made at the beginning and maintained over time based on personal preference. Consequently, they should be very proficient in typing. Recently, many websites propose that the style of typing reflects personality characteristics [53]. As for DIs, based on the above, we can assume that they continue to use the same type of interaction with the keyboard that they used previously on the typewriter or computer (typing with one finger), in which they are very skilled.

The analysis of the typing sequence and of the kinematics highlighted the presence of variables that could not be predicted a priori during the choice of the stimuli to be used due to the absence of information in the literature. This led us to use a few stimuli in the hope of identifying the critical variables. Based on the present data, an in-depth examination of the many possibilities supplied by the different applications to identify specific digital affordance appears necessary. Furthermore, there emerges the need to control the number of typing actions in the stimuli that are presented. Future research will be able to manipulate these variables and test their impact on performance.

The limitations of this study are mainly due to the fact that there is no literature on the present topic. There are no data to assess typing ability, neither on the kinematic characteristics of goal-based typing, nor on the cortical representation of different typing styles. However, the results, based on a large sample of participants, not only confirmed the expected results—better performance as age decreased—but also offered new ideas for investigating the execution and recognition of digital intentions. In fact, confirming what happens in the physical world, they suggest that typing ability has a fundamental role in this process. They further suggest that this process exploits not only some kinematic differences characterizing the different digital intentions—higher Peak Velocity during content generation—but also the recognition of specific digital affordances (i.e., the presence of swipe). However, given the lack of data in the literature, this interpretation of the results is based on hypotheses and on preliminary data relating to the functional and kinematic characteristics of the stimuli. To confirm the results and interpretation, it will be necessary to repeat the experiment with more stimuli and trials. This will only be possible after studying the differences in digital affordance and kinematics between the actions of generating and consuming digital content. It will also be necessary to define new indices to identify the level of competence in touchscreen typing. Altogether, this knowledge will allow future studies to use more refined experimental procedures, such as Signal Detection Theory paradigms, and highlight the many variables that can affect performance.

We hope that this study will initiate new lines of research that will fill the gaps in the literature, paving the way for defining sensorimotor representations of digital actions in the context of embodied cognition. The study of this topic will become increasingly necessary, given the fast pace of activities performed almost exclusively online. This knowledge will be needed to fight digital exclusion at different levels. Knowledge of the neural bases of interaction with digital devices will also make it possible to identify and intervene in the case of specific deficits. For instance, as is the case in the physical world, it can be hypothesized that lesions to different and specific cortical areas may induce selective symptoms for the generation or consumption of digital content [31]. Currently, there is only anecdotal evidence in this regard, given the lack of suitable neuropsychological tools for their diagnosis.

## 5. Conclusions

In conclusion, the results of the present study reveal, for the first time, that by watching an agent’s hands typing on a smartphone an observer can determine whether the agent is typing to obtain information or to share it with others. The probability of answering accurately varies based on the age and typing style of the observer. Specifically, only DIs who declared that they type with both thumbs—i.e., the typing style mostly utilized by DNs, and to a lesser extent by DIs—performed worse than the others. We propose that the recognition process is based on identifying small variations in the agent’s movement, a skill that grows with the ability to type. We hypothesize that DI individuals who type with two thumbs are less skilled in typing than the other groups, and so behave as less expert observers. This preliminary work aims to serve as a starting point for further research on sensorimotor representations of actions on touchscreens. This knowledge is critical for defining digital competence, a necessary index for any intervention aimed at overcoming digital exclusion.

## Figures and Tables

**Figure 1 brainsci-13-01418-f001:**
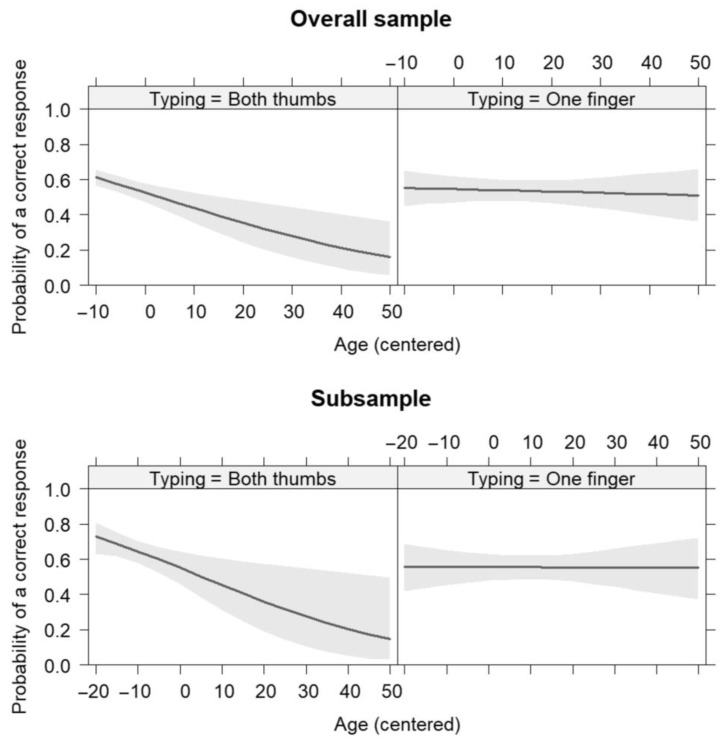
Probability of a correct response considering age as a continuous variable. “Overall sample” graphs show the results of the complete sample. “Subsample” graphs show the results of the Subsample where typing modality was paired and matched across sexes. In both cases, the graphs on the left (“Typing = Both thumbs”) show the results of participants who typed using both thumbs, and the graphs on the right (“Typing = One finger”) of those who typed using one finger. “Age (centered)” indicates that the value 0 in abscissa coincides with the average age of the sample. Gray areas represent standard error.

**Figure 2 brainsci-13-01418-f002:**
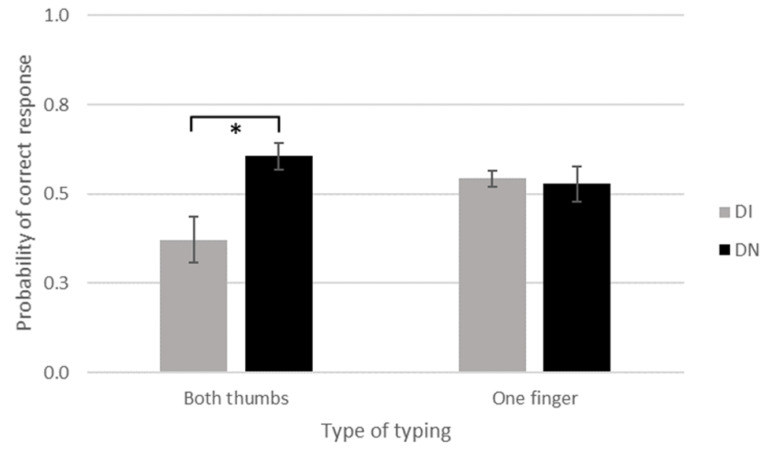
Probability of a correct response considering age as a discrete variable. Mean values are reported for Digital Immigrants (DI; grey bars) and Digital Natives (DN; black bars), reporting to type with both thumbs or with one finger. Thin lines above histograms indicate standard error of the mean. * indicates the presence of a statistically significant difference.

**Figure 3 brainsci-13-01418-f003:**
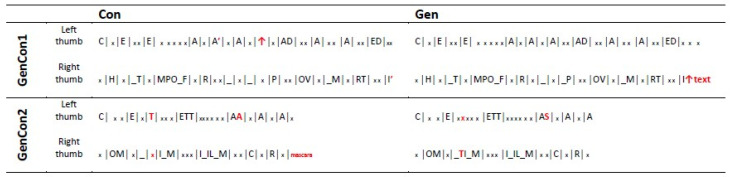
Typing sequence for each thumb in each condition. This figure schematically illustrates the typing sequence for each thumb present in the GenCon1 (Con = Google search window; Gen = WhatsApp message window) and GenCon2 (Con = YouTube search window; Gen = WhatsApp message window) videos. Capital letters indicate the keys pressed by the thumb. The small xs indicate the keys pressed by the other thumb. Discrepancies between Con and Gen conditions are shown in red. Underscore indicates the space bar. The symbol ↑ indicates the selection of the caps key. The symbol ↑_text_ indicates the pressing of the function key and the selection of the cued text. The word _mascara_ indicates the selection of the cued correct word (see text for details).

**Figure 4 brainsci-13-01418-f004:**
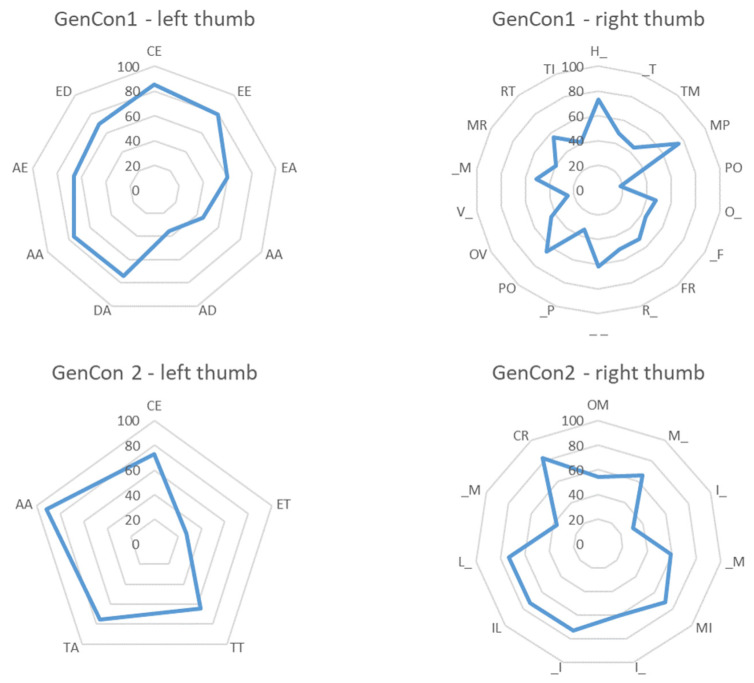
Percentage of overlap of speed data between conditions. Radar graphs reporting the percentage of overlap between condition Gen and Con for each basic typing action (GenCon1, upper panels; GenCon2, lower panels) and both thumbs (left thumb, left panels; right thumb, right panels).

**Table 1 brainsci-13-01418-t001:** Generalized mixed effects models (GLMMs) results relative to the Overall sample and to the Subsample.

	Response Variable: Correct/Incorrect		
Models	β	*p* Value	OR (CI)
**Overall**			
Video pair	0.85	**<0.001**	2.35 (1.64–3.37)
Age	−0.02	**<0.01**	0.98 (0.97–0.99)
Typing	−0.08	0.68	0.92 (0.64–1.34)
Video pair × Age	0.02	0.17	1.02 (0.99–1.04)
Video pair × Typing	0.63	0.08	1.88 (0.93–3.82)
Age × Typing	−0.03	**<0.01**	0.97 (0.94–0.99)
Video pair × Age × Typing	−0.004	0.86	1.00 (0.95–1.04)
**Subsample**			
Video pair	0.90	**<0.001**	2.47 (1.55–3.93)
Age	−0.02	**0.02**	0.98 (0.96–0.99)
Typing	−0.02	0.94	0.98 (0.64–1.60)
Video pair × Age	−0.007	0.63	0.99 (0.96–1.02)
Video pair × Typing	0.42	0.37	1.52 (0.62–3.73)
Age × Typing	−0.04	**0.02**	0.96 (0.93–0.99)
Video pair × Age × Typing	−0.04	0.20	0.96 (0.90–1.02)

## Data Availability

The datasets generated during the current study are available in the OSF repository, https://osf.io/3md7v/?view_only=cdcd4affcdc8443a8113f3f021f418f9 (accessed on 2 December 2022).

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
