# Peer review of "Digital Intentions in the Fingers: I Know What You Are Doing with Your Smartphone"

_brainsci, 2023, doi:10.3390/brainsci13101418_

Round 1
Reviewer 1 Report
The manuscript titled "Digital intentions in the fingers: I know what you are doing with your smartphone", is an interesting study that delves into comprehending the activities performed online and draws the difference in digital patterns between content generation and content consumption.
It would be nice to have more evidence through varied stimuli and a larger number of trials for generalisation. Also, the direct application and purpose of such a study design are still unclear.
Overall, a decent study to understand digital intentions.
Author Response
Please see the attachement.

Reviewer 2 Report
This is an excellent paper opening a new research area. I would have liked to see some potential approaches on the computational aspects of this competence. Of course this is mostly a psychology paper, but it would be nice to have a paragraph describing what the authors believe to be the underlying computations
no
Author Response
Please see the attachement.

Reviewer 3 Report
lines 51-54- Affordances are possible actions that emerge from object/ambient detected properties and actor capabilities. Please, review concept, it may help you to better understand your results, and to be more in line with your conceptual approach (e.g., Stoffregen, T. A. (2003). Affordances as Properties of the Animal–Environment System. Ecological Psychology, 15(2), 115–134.). Notice, for exemple, lines 259-263 (intrinsic constraints; also, lines 465-471), and, lines 263-266 (extrinsic constraints; also, lines 489-516); and, for both constraints, an affordance emerges (lines 283-286; also line 564).
line 170- maybe the term "stimulus" is not adequate in the theoretical framework assumed. we suggest "stimulation" (see, Gibson's direct perception theory for more details)
lines 372-377- Include references to sustain these parameters (any classical study of reaching and grasping will be adequate)
lines 390-395- and so, right-handed prevalence in humans, and some genetic theories about, become completly unsupported (and much probably wrong)
line 417 (and similar events)- consider "stimulation" instead of "stimulus", namely because the "stimulation" concept allows to consider that there is information during sensory and perceptual detection, in line with cognitive embodied theory
General discussion- forgot your theory? what about discuss it, based on results obtained
Author Response
Please see the attachement
